# Zinc Oxide Nanoparticles: Physiological and Biochemical Responses in Barley (*Hordeum vulgare* L.)

**DOI:** 10.3390/plants11202759

**Published:** 2022-10-18

**Authors:** Marina Voloshina, Vishnu D. Rajput, Tatiana Minkina, Evgeniy Vechkanov, Saglara Mandzhieva, Mahmoud Mazarji, Ella Churyukina, Andrey Plotnikov, Maria Krepakova, Ming Hung Wong

**Affiliations:** 1Academy of Biology and Biotechnology, Southern Federal University, 344090 Rostov-on-Don, Russia; 2Division for Allergic and Autoimmune Diseases, Rostov State Medical University, 344000 Rostov-on-Don, Russia; 3Consortium on Health, Environment, Education, and Research (CHEER), and Department of Science and Environmental Studies, The Education University of Hong Kong, Tai Po, Hong Kong 999077, China

**Keywords:** malonic dialdehyde, superoxide dismutase, catalase, reactive oxygen species, antioxidant enzymes, metabolic changes, stress marker, ZnO NPs, pollution

## Abstract

This work aimed to study the toxic implications of zinc oxide nanoparticles (ZnO NPs) on the physio-biochemical responses of spring barley (*Hordeum sativum* L.). The experiments were designed in a hydroponic system, and *H. sativum* was treated with two concentrations of ZnO NPs, namely 300 and 2000 mg/L. The findings demonstrated that ZnO NPs prevent the growth of *H. sativum* through the modulation of the degree of oxidative stress and the metabolism of antioxidant enzymes. The results showed increased malondialdehyde (MDA) by 1.17- and 1.69-fold, proline by 1.03- and 1.09-fold, and catalase (CAT) by 1.4- and 1.6-fold in shoots for ZnO NPs at 300 and 2000 mg/L, respectively. The activity of superoxide dismutase (SOD) increased by 2 and 3.3 times, ascorbate peroxidase (APOX) by 1.2 and 1.3 times, glutathione-s-transferase (GST) by 1.2 and 2.5 times, and glutathione reductase (GR) by 1.8 and 1.3 times in roots at 300 and 2000 mg/L, respectively. However, the level of δ-aminolevulinic acid (ALA) decreased by 1.4 and 1.3 times in roots and by 1.1 times in both treatments (nano-300 and nano-2000), respectively, indicating changes in the chlorophyll metabolic pathway. The outcomes can be utilized to create a plan of action for plants to withstand the stress brought on by the presence of NPs.

## 1. Introduction

The widespread inclusion of nanotechnology, especially nanoparticles (NPs), in the production of various products leads to their distribution in the environment (water, soil, air) and interaction with living organisms (bacteria, animals, plants) [1,2]. Due to their small size (1–100 nm) and high surface-to-volume ratio, NPs have unique chemical and physical characteristics compared to microparticles of the same metals [3,4]. They can pass through plant cell membranes, transport substances into cells, and integrate into metabolic pathways [5,6]. However, the introduction of NPs entails several physiological changes in plants [7].

The most common consequence of NP exposure is the development of a cascade of reactions leading to plants’ oxidative stress. This is due to the formation of an increased concentration of reactive oxygen species (ROS), which include superoxide anion (O_2_-˙), hydrogen superoxide (OH˙), peroxide oxygen (H_2_O_2_), and singlet oxygen (‘O_2_). ROS are involved in the development of oxidative stress by activating various processes in the plant, one of which is lipid peroxidation (LPO) [8]. The accumulation of LPO products occurs as a result of the oxidation of lipid membranes, proteins, and amino acids. The main product of LPO is malondialdehyde (MDA). This is one of the essential mechanisms of the plant aging process [9].

A similar response in many plant species subjected to various abiotic stresses is noted for the accumulation of metabolites such as proline. The cell uses proline to regulate the osmotic pressure between the cytoplasm and vacuole and to counteract the toxic effects of ROS [10,11]. Currently, the mechanism of release and accumulation of proline in plants or their parts subjected to stress is poorly understood [10,12]. It is hypothesized that this decrease in mitochondrial electron transport system activity would lead to the accumulation of NaDH and H^+^. At high proline concentrations, the reduction in δ-aminolevulinic acid (ALA) was observed in plants [12,13]. ALA strengthens the antioxidant defense mechanism that shields the plant membrane from ROS and increases plant growth and yield at low concentrations. It also enhances plants’ stress resistance to heavy metal (HM) accumulation, especially when they enter in the form of NPs [14]. However, the regulatory mechanisms behind stress resistance are not yet well known. There are few references in the scientific literature describing the role of ALA in regulating plant growth and repairing damage caused by abiotic stresses [15].

Glutathione (GSH), along with ascorbate, is a crucial non-enzymatic antioxidant included in the ascorbate–glutathione detoxification cycle for H_2_O_2_ and metal chelation. GSH is a vital redox buffer of plant activity, and its constituent part is the level of synthesis and the degradation of the balance that plants have under growing conditions. However, attention should be paid to the interaction of GSH with other components of the plant antioxidant system. Thus, it is inherently combined with the activities of GSH, glutathione reductase (GR), glutathione S-transferases (GST), and ascorbate peroxidase APOX [16]. Superoxide dismutase (SOD) catalyzes the dismutation of superoxide ions to H_2_O_2_ and O_2_ and OH radical production from hydrogen peroxide, dramatically reducing superoxide-mediated toxicity. In addition to regulating ROS signaling, it also controls ROS damage [17]. The direct or indirect effects of NPs on the SOD gene expression or ROS level may be caused by an increase or reduction in SOD activity [11].

ZnO NPs have a favorable effect on plant growth and development in modest quantities. Still, they can have a harmful impact by displacing other ions of elements from protein-binding sites, which causes the generation of ROS in reaction to stressful situations [18,19,20]. Although oxidative stress caused by exposure to NPs has been seen in various animal tissues, nothing is known about how it affects plants [21]. The chemical composition, size, and reactivity of NPs determine their toxic effects [22,23]. This is also because NPs of metals are absorbed by plants 15–20 times more than their larger particles [23,24]. It is well-explored that Zn ions are part of the transcription factors known as zinc fingers that control cell proliferation and differentiation [25].

The study of the physiological and biochemical processes of the entry of microelements into plants, including HMs, especially in nano-dispersed forms, remains relevant due to the rapid development of nanotechnology [23]. Nowadays, very little information on the mechanisms of plant adaptation to an excess of ZnO NPs is known. It was assumed that the excessive accumulation of Zn dissolute from ZnO NPs in plants provokes ROS production, leading to oxidative stress and a reduction in plant growth and development. This **study aimed** to examine the physiological and biochemical reactions of *H. sativum* cultivated in hydroponic systems after the addition of ZnO NPs in low and high dosages.

## 2. Results

### 2.1. Structural and Morphology Characterization of ZnO Nanoparticles

The surface of the sample under study consists of only one ZnO crystalline phase with a hexagonal symmetry system, as shown in Figure 1a. The X-ray phase structural analysis (XRD) data show that the sample contains no other substance’s crystals, indicating the high purity of the analyzed NPs. A small proportion of particles are in a scattered form and particle sizes vary from 10 to 50 nm. The most significant frequency of the studied particle sizes lies from 15 to 30 nm (Figure 1b). Dynamic light scattering (DLS) measurements were performed to measure the zeta potential of the ZnO NPs of both concentrations (data not shown). The zeta potential readings of −22 mV for 300 mg/L and −6.5 mV for 2000 mg/L indicated considerable stability. This means that the prepared colloidal system containing ZnO NPs possessed excellent stability and remained intact without agglomeration during the experiment [26].

The FTIR spectrum of the ZnO NPs is presented in Figure 2. The broad absorption band at 3447 cm^–1^ was attributed to the normal polymeric O–H stretching vibration of H_2_O in the ZnO sample. The band centered at 1620 cm^–1^ is the vibrational bending mode of the hydroxyl group of the chemisorbed and/or physisorbed water molecules. More importantly, the peak associated with the vibration of Zn-O appeared in the spectrum at around 500 cm^–1^.

### 2.2. Morphobiometric Parameters and Accumulation of Zn in Plant Tissues

The visual observation of plants showed growth, leaf color, and root morphology changes compared to the well-developed root system in the control plants (Figure 3). The length of the leaves was 11.3 ± 1.3 cm and the length of the roots was 12.1 ± 1.1 in the control (Figure 4a). The results of the control and contaminated *H. sativum* showed a decrease in the total length by 2.0–3.3 times (Figure 4a) and dry weight by 1.2–2.0 times (Figure 4b), depending on the dose of NP application. The shoot height and root length decreased by 1.4 and 1.8 times and 1.8 and 2.5 times compared with the control, respectively (Figure 4). In the variant nano-2000, the increase in the content of Zn in the aerial part was 2.0 times compared to the variant nano-300 (Figure 4c).

The germination rate of *H. sativum* seeds treated with ZnO NPs decreased by 17% compared to the control. After growing for 2 weeks, the seedlings showed a decrease in root length by 59.3 and 71.7% and shoot height by 29.8 and 42.9% for the nano-300 and nano-2000 experimental variants, respectively.

### 2.3. Effects of ZnO NPs on Biochemical Indices

#### 2.3.1. Malonic Dialdehyde

The average content of MDA in the plant shoots of the control group was 0.031 ± 0.003 and 0.015 ± 0.002 mM/g DW, accordingly (Figure 5a). The shoot MDA concentration was increased by 1.17 times in the variants of nano-300 and 1.69 times in nano-2000 compared with the control (Figure 5a). In the roots, a dose-dependent increase in MDA was also observed by 1.19 and 1.56 times in nano-300 and nano-2000, respectively (Figure 5a).

#### 2.3.2. Protein

The control variant’s protein content was 0.88 ± 0.015 and 0.323 ± 0.001 mg/g DW in the shoots and roots, respectively (Figure 6b,d). Both nano-300 and nano-2000 groups had a decrease in shoot and root protein contents, depending on the NP doses applied. There was a reduction in protein contents in shoots by 1.3 and 1.7 times in nano-300 and nano-2000 groups, respectively (Figure 5b). Furthermore, plant shoots showed a concentration-dependency trend of 1.2 and 1.4 times at doses of 300 and 2000 mg/kg, respectively (Figure 5b).

#### 2.3.3. δ-Aminolevulinic Acid

The proline accumulation in the experiment without NPs was 4.8 ± 0.43 and 2.7 ± 0.33 nmol/g DW in shoots and roots, respectively. The δ-ALA in shoots was increased by 1.4 and 1.3 times and in roots by 1.9 and 1.13 times in nano-300 and nano-2000, respectively (Figure 5c).

#### 2.3.4. Proline

The content of proline in the *H. sativum* control was noted in shoots at 0.65 ± 0.07 mM/g DW and in roots at 0.58 ± 0.07 mM/g DW) (Figure 5d). The proline in roots was increases by 1.03 and 1.09 times in shoot proline and 1.1 and 1.69 times in the nano-300 and nano-2000 groups than in the control (Figure 5d).

#### 2.3.5. Catalase

In the control, CAT content averaged 81.50 ± 5.40 nMol/mg of protein in the shoot and 36.52 ± 5.27 nmol/mg of protein in the roots (Figure 6a). The shoot CAT activity increased by 1.4 and 1.6 times in the nano-300 and nano-2000 groups. A similar pattern was observed in the root CAT activity, with an increase of 1.3 and 1.6 times in the two groups.

#### 2.3.6. Ascorbate Peroxidase

The ascorbate peroxidase activity (APOX) in the control samples was 0.98 ± 0.1 and 1.73 ± 0.2 mM/mg of protein in shoots and roots, respectively (Figure 6b). There are significant differences from previous studies conducted at different growth phases. A significant increase in enzyme activity was obtained by 1.2 and 1.3 times in nano-300 and nano-2000 groups. In shoots, the changes were insignificant, from 1 to 9%.

#### 2.3.7. Superoxide Dismutase

The shoot and root SOD contents were 11.7 ± 1.90 and 28.30 ± 2.5 U/mg of protein in the control group (Figure 6c). Compared with the control, the decrease in shoot SOD by 1.1 and 1.1 times were recorded in nano-300 and nano-2000 groups. The root SOD contents were raised by 2 and 3.3 times in the two groups.

#### 2.3.8. Glutathione and Glutathione Reductase

The glutathione reductase activity increased in the roots and shoots by the presence of the nano-300 and nano-2000 groups. In the control samples, the enzyme activity was 25.4 ± 4.41 and 89.11 ± 7.92 U/g of protein (Figure 5b and Figure 6e). An increase in GR activity was accompanied by an insignificant increase in reduced glutathione content in all treatment groups in nano-300 and nano-2000 groups, respectively. In the nano-300 group, the most significant increase in enzyme activity was recorded in roots by 1.84 times.

#### 2.3.9. Glutathione-s-Transferase

Glutathione-s-transferase (GST) activity in the control sample was 92.93 ± 19.02 and 204.50 ± 27.50 U/g of protein in shoots and roots, respectively (Figure 6f). A dose-dependent increase in GST by 1.2 and 2.6 times was observed in the nano-300 and nano-2000 groups in the roots. In the nano-300 treatment, there was a slight decrease of 1.1 times in the stems. On the contrary, an increase of 1.1 times was observed in the shoots.

## 3. Discussion

A high dose of ZnO NPs (2000 mg/L) increases Zn content two-fold in plant shoots and affects biochemical indicators. However, the root is the first plant organ exposed to ZnO NPs. Damage and the accumulation of Zn were noted to be higher (10-fold higher) than in the shoots. A high accumulation of ZnO NPs affected the morphometric parameters (root and shoot length, biomass) and biochemical indices (malonic dialdehyde, total protein, δ-aminolevulinic acid, and proline levels) of *H. sativum* grown in the hydroponic system. The hydroponic system allowed us to simplify the experiment for the most expressive results by controlling plant growth indices [27,28,29].

The DLS and hydrodynamic observations noted the high stability of ZnO NP suspensions. The ZnO NPs used contain no crystals of other substances, as demonstrated by XRD analysis, which may indicate the absence of impurities during the analysis. FTIR showed the main peak related to the Zn−O stretching vibration at 510 cm^−1^. SEM observations determined that the most considerable frequency of the sizes of the studied particles lies in the range from 15 to 40 nm. These characteristics of ZnO NPs ensured the stability and accuracy of the experiment performed in the hydroponic system. The primary response of plants to high Zn concentrations is the inhibition of root growth, thickening, and the disruption of cell division. Cell viability was also observed in other studies [30,31]. The toxicity of Zn^2+^ in hydroponic solutions may be related to the increased availability of metal ions to plant roots [18,32].

The symplastic pathway transports Zn to the aerial part of plant tissues. In some cases, NPs could interfere with the transport of microelements and water in the stems and shoots of plants. During the translocation, Zn competes with Cu, Fe, and Mg and impacts the kinetics of redox enzymes [33]. The Zn has a strong chemical similarity to both Fe and Mg and can replace these two metal ions in the active sites of enzymes, thus interfering with cellular functions [34]. An enhancement was noted by the addition of low concentrations of ZnO NPs in redox-inactive metal that is used as an agent for pool activation by the enzyme of the plant antioxidant system [35]. However, it also disturbs the metabolic balance of the plant, as Zn inhibits several cellular enzymatic reactions [36]. Increases in ROS result in the enhancement of free radicals’ generation, which leads to an increase in the LPO process. As a result, MDA accumulation reduces plants’ physiological activities, growth, and development [37,38]. The highest MDA content correlates with the observed physiological inhibition of *H. sativum*. In addition to lipid peroxidation, ROS production is more toxic to plant cells than indirect effects [39].

Proline accumulation is a biochemical response of plant cells to stress conditions [40]. Under stress conditions, there is a decrease in the reserves of proline used by the plant to neutralize ROS. It is believed that proline’s protective effect is related to its ability to detoxify ROS [10], inhibit lipid peroxidation [41], and act as an osmoprotectant in the case of metal toxicity [42]. Several studies have shown that exposure to metals in micro-and nano-form has revealed changes in proline content in various plant species [24,43,44]. Increasing total soluble sugars and protein has been linked to metals, and this impact was more pronounced in shoots than in roots [45]. Perhaps this explains our experiment’s more significant increase in proline in *H. sativum* shoots.

Since both δ-ALA and proline are synthesized from a common precursor, glutamic acid, their role as antistressors can be interchangeable in vegetation periods and various plant conditions such as salinity, drought, and oxidative stress [2,42,46]. The present study showed an increase in δ-ALA, possibly due to barley’s enzymatic changes and ROS production. This suggested that δ-ALA metabolism was rebuilt from the path of synthesis of chlorophyll and heme to the path of proline synthesis, thereby increasing stress resistance.

Various growth conditions affect the metabolic activity of cells based on their total protein content. Protein degradation is closely related to the stages of plant vegetation [47]. Intracellular proteins are oxidized when exposed to free radicals and, as a result, are destroyed by proteolytic enzymes [48]. In the present study, root and shoot protein contents were reduced depending on the amount of NPs applied. In another experiment, ZnO NPs (8 mg/L) increased the protein content by 45% in tomato shoots compared with their control [49]. It is generally known that Zn is essential for maintaining the balance of ROS production, which maintains the stability of proteins [50]. However, excess Zn in plant tissues might reduce protein content in plant tissues, which was noted in the present study.

The functional activities of cellular membranes were noted to be sustained by redox enzymes. The redox enzymes could alleviate stresses in plants imposed by excessive ZnO NP accumulation. The antioxidant activity of enzymes in plants exposed to the toxic effects of ZnO NPs varies greatly depending on the plant species, their concentration, and the duration of exposure. The comparative activity of antioxidant enzymes may lead the plant to initiate a strong antioxidant response depending on the development of ZnO NPs. Studies have shown an increase in the level of low-molecular-weight antioxidants.

Proline and GSH are associated with metal chelation, reducing their toxic effect. Moreover, low concentrations of ZnO NPs have a stimulating effect due to the accumulation of osmolytes in plant organs, making it possible to alleviate oxidative stress [17]. The main indicator of GST is the conjugation of xenobiotics and secondary metabolites with the formation of low-toxic peptide derivatives. There is a sharp increase in the activity of GST and GR in the barley roots. However, the increase in leaf enzyme activity was only recorded at the dosage of nano-2000. This may be due to a slight increase in GSH in the roots and shoots of *H. vulgare*, as it is a substrate for reducing glutathione cycle enzymes. The greatest response of enzymes to the toxic effect of ZnO was observed in the roots during the entire ASC–GSH cycle.

The ZnO NPs are a cofactor of SOD, so in small quantities, they stimulate plants. However, high concentrations accumulated in plants overload cells and disrupt Zn homeostasis [51,52], leading to a loss of ROS production. One of the primary antioxidant enzymes, SOD, detoxifies super-radicals to O_2_ and H_2_O_2_ and counteracts the damaging effects of ROS [53]. However, H_2_O_2_ is also found in plant tissue, one of the agents of ROS. There was an increase in lipid peroxidation and CAT activity after treatment with ZnO NPs [54].

Their effective function is to detoxify H_2_O_2_ to H_2_O and O_2_ in the cytoplasm or organelles of the cell. Currently, a dose-dependent increase in the concentration of APOX and CAT in *H. vulgare* plants is observed. However, a pronounced antioxidant response is observed in APOX, recorded in the roots, and in CAT in the shoots, manifesting itself in the multidirectional leveling of the stress of enzyme fibers.

## 4. Materials and Methods

### 4.1. Preparation, Characterization, and Application of Nanoparticles

For the preparation of the required concentration, ZnO NPs (size: <50 nm, purity: >97%, CAS-No. 1314-13-2; Sigma–Aldrich, St. Louis, MO, USA) were poured into double-distilled water. A well-mixed dispersion of the NPs was achieved by shaking and ultrasonically stabilizing the solution prior to use in the experiments. This step minimized aggregation and agglomeration. An XRD was used to determine the crystal structure of the ZnO NPs. The functional groups were measured on an FSM-1202 spectrometer in transmission mode with a DTGS detector.

### 4.2. Plant Growth Condition and Sample Collection for Morpho-Physiological and Biochemical Indices

The seeds of *H. sativum* were used as widely cultivated in the Rostov region of Russia as a valuable food and fodder crop. Twenty-five seeds were distributed evenly over filter paper in Petri plates after being visually inspected for any damage and germinated at 28 °C by adding 5 mL of distilled water. The seedlings were moved into plastic containers (100 × 60 × 50 mm) with 50 mL of water without NPs (ZnO NPs; 00 mg/L) and with NPs (ZnO NPs; 300 and 2000 mg/L) and labeled as control, nano-300, and nano-2000 after successful germination. The doses of NPs were chosen to take into account the concentrations of Zn in the Rostov region’s soils. The seedlings were grown for 14 days under hydroponic conditions with constant stirring to avoid the aggregation of NPs.

Following the experiment, samples of the roots and shoots were taken, cleaned with distilled water, and kept at −80 °C for later examination. After homogenizing plant tissues in extraction solution and centrifugation at 12,000× *g*, 4 °C for 6 min, the supernatant was used to observe biochemical indices (7 replicates) using Beckman Coulter DU 800 spectrophotometers.

### 4.3. Determination of Zn in Barley

The Zn concentration in plant tissues was measured using the combustion method, which involved burning the tissues at 450 °C. The ash was dissolved in 5 mL of 20% HCl and then filtered through 0.45 m Whatman filter paper from the air-dried 1 g samples. The analysis was made using an atomic absorption spectrophotometer (KVANT 2-AT, Kortec Ltd., Moscow, Russia) with a wavelength range of 213.9 nm at room temperature.

### 4.4. Morphobiometric Analysis

Morphobiometric indices were determined by measuring the lengths of the roots and shoots. The germination energy was calculated according to the following formula [55].
(1)$$\mathrm{Relative}\;\mathrm{seed}\;\mathrm{germination}\;\mathrm{inhibition}\;\left(\%\right)=\frac{{\mathrm{germination}}_{\mathrm{control}\;}{\text{-}\;\mathrm{germination}}_{\mathrm{treat}}}{{\mathrm{germination}\;}_{\mathrm{control}}}\;\times \;100$$

### 4.5. Biochemical Indicators

#### 4.5.1. Malonic Dialdehyde

MDA was measured to determine the extent of lipid peroxidation (LPO). The determination of MDA was based on the reaction of MDA with thiobarbituric acid (TBA) [56]. Measurements were performed on a spectrophotometer at λ = 532 nm and λ = 600 nm to correct for nonspecific absorption [57]. The MDA content was expressed in mM/g DW. The calculation was carried out using the Beer–Lambert equation.

#### 4.5.2. Proline

The proline content was analyzed by a modified method by Bates et al. [58], based on the colored product formed after proline’s interaction with the ninhydrin reagent. The amount of proline was determined by the color intensity of the solution on a spectrophotometer, λ = 520 nm. The amount of proline is expressed in U/mg of protein. The calculation was carried out using a standard curve.

#### 4.5.3. δ-Aminolevulinic Acid

The δ-aminolevulinic acid (ALA) content was based on the reaction of porphobilinogen treated with a modified Ehrlich reagent [59]. The optical density of the solution at λ = 553 nm. The amount of ALA was calculated using a molar extinction coefficient of 6104 xM^–1^ xcm^– 1^. The calculation was carried out using the Beer–Lambert equation.

#### 4.5.4. Protein

Protein concentration was determined by the Bradford method [60]. This method was based on the shift of the absorption spectrum of Coomassie Blue towards λ = 595 nm, directly proportional to the concentration of the protein contained in the solution. A buffer solution was used as a control. The calculation was carried out using a standard curve.

#### 4.5.5. Superoxide Dismutase (EC 1.15.1.1)

The activity of superoxide dismutase (SOD) was based on the inhibition of the reduction in nitroblue tetrazolium (NBT) during the autooxidation of adrenaline to adrenochrome in an alkaline medium under conditions of the generation of the superoxide anion radical. The reaction mixture consisted of 66 mM sodium phosphate buffer (pH 7.4), 6.11 mM nitro bluezolium chloride (NBT), 1 mM 1,4-dithiothreitol, 0.5 mM phenylmethylsulfonyl fluoride, and 1 mg polyvinylpyrrolidone. Measurements were made spectrophotometrically (Beckman coulter DU 800, Fullerton, CA, USA) using the maximum wavelength of 540 nm [61]. Enzyme activity was stated in mM/mg of protein. The calculation was carried out using a standard curve.

#### 4.5.6. Catalase (EC 1.11.1.6)

In the presence of catalase and a biological substrate, catalase activity (CAT) was measured by analyzing the H_2_O_2_ complex with ammonium molybdate [43]. The measurements were carried out spectrophotometrically λ = 410 nm [62], with the enzyme activity expressed in nmol/mg of protein. The calculation was carried out using a standard curve.

#### 4.5.7. Glutathione Reductase (EC 1.8.1.7)

The activity of glutathione reductase (GR) was based on the reduction in oxidized glutathione in the presence of nicotinamide adenine dinucleotide phosphate (NADPH) λ = 340 nm and with the reaction mixture containing plant extract, 0.2 M sodium phosphate buffer (pH 7.4), 0.1 M KCl, 8 mM ethylenediaminetetraacetic acid (EDTA), 2 mM NADPH, and 8 mM glutathione disulfide, spectrophotometrically [63]. Enzyme activity was measured as mM/mg of protein.

#### 4.5.8. Ascorbate Peroxidase (EC 1.11.1.11)

The activity of ascorbate peroxidase (APOX) was determined by the rate of the decomposition of H_2_O_2_ by APOX of the test sample, resulting in the formation of water and dehydroascorbate [64]. The reaction mixture (3 mL) contains 50 mM sodium phosphate buffer (pH 7.0), 0.5 mM ascorbic acid, 0.1 mM EDTA, 0.1 mM H_2_O_2_, and 0.1 mL of plant extract. The reaction was started by adding H_2_O_2_, and then the change in optical density was measured at 290 nm for 3 min. Then, the change in optical density was measured at λ = 290 nm for 3 min. Enzyme activity is expressed as mM/mg of protein. The calculation was carried out using a standard curve.

#### 4.5.9. Glutathione-S-Transferase (EC2.5.1.18)

The activity of glutathione-S-transferase (GST) was determined by the rate of the formation of glutathione-S conjugates between reduced glutathione (GSH) and 1-chloro-2,4-dinitrobenzene (CDNB) at λ = 340 nm [65]. The supernatant (0.1 mL) was added to the reaction mixture containing 2.5 mL of 0.1 M potassium phosphate buffer (pH 6.5) and 0.2 mL of 0.015 M GSH. The reaction was initiated by adding 0.2 mL of 0.015 M CDNB. The glutathione-S-transferase activity is expressed as mM/min*mg of protein. The calculation was carried out using a standard curve.

#### 4.5.10. Glutathione Analysis

The definition was based on the interaction of reduced glutathione (GSH) with DTNBA to form a yellow-colored 2-nitro5-thiobenzoate anion. The increase in the concentration of the yellow anion during this reaction was recorded spectrophotometrically λ = 412 nm [66]. Determining the total content of glutathione and reduced GSH and oxidized (GSSG) forms of glutathione was carried out spectrophotometrically [67]. The total glutathione content in the samples was measured (color reaction) due to forming a complex of 5,5′-dithiobis-2-nitrobenzoic acid (DTNBA) and GSH. To evaluate the content of GSSG, 2-vinylpyridine was used, which binds to GSH. The thionitrophenyl anion content was calculated using an extinction coefficient of 12.7*103M/cm. The glutathione content was expressed in μM/g of protein. The calculation was carried out using the Beer–Lambert equation.

### 4.6. Statistical Analysis

Statistical analysis was performed with SPSS-19 and Microsoft Excel 2016. The Mann–Whitney U test was used to establish statistical significance. The coefficient of variation was <10%. The significance of the differences among replicates was defined at *p* < 0.05. The bars on the figures indicate the standard deviation level.

## 5. Conclusions

For the first time on the plants of *H. sativum* L., physiological and biochemical reactions ensure monocot plants’ resistance to increased ZnO NP contents. ZnO NP contamination negatively affected the growth and development of *H. sativum*. The concentrations introduced (300 mg/L and 2000 mg/L ZnO NPs) had a toxic dose-dependent impact on plants’ morpho-biochemical indices. This is confirmed by the accumulation of MDA in all plant organs, which provokes metabolic changes and enhances membrane permeability and further destruction. The proline analysis of the plant’s aerial part revealed the accumulation of a low-molecular-weight antioxidant. The activity of antioxidant enzymes was manifested depending on the plant organism. Thus, in the aerial part, the sharpest increase was observed in CAT. In the core, SOD, APOX, GR, and GST activity was most pronounced, associated with the primary contact with ZnO NPs. The increase in GSH is associated with the activation of the ASC–GSH cycle due to ZnO NPs. However, the level of δ-aminolevulinic acid in the plant decreased, indicating metabolic changes in the synthesis of chlorophyll and heme. A decrease in protein in shoots and roots was observed. These changes are associated with the accumulation of Zn in the organs of *H. sativum*. Various environmental factors could impact the bioavailability of ZnO NPs. Therefore, prolonged in situ studies are needed to explore more insights into biochemical indications, activities, and modifications and the role in the activation of plant defiance mechanisms to adapt to stress conditions, especially in *H. sativum*.

## Figures and Tables

**Figure 1 plants-11-02759-f001:**
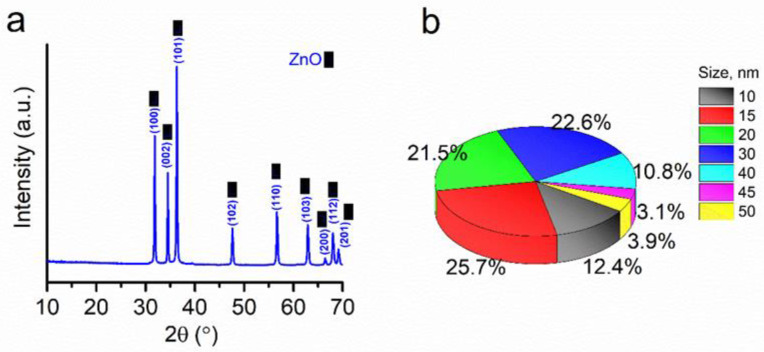
ZnO NPs. (**a**)—XRD pattern and (**b**)—particle size distribution.

**Figure 2 plants-11-02759-f002:**
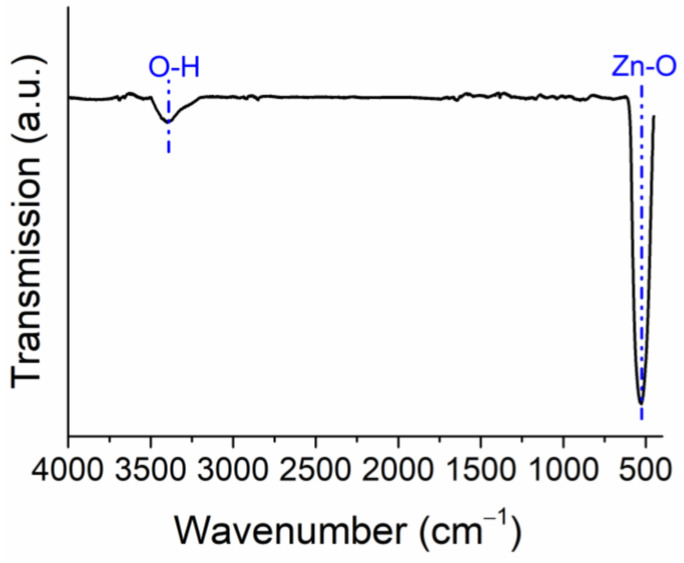
FTIR spectrum of ZnO NPs.

**Figure 3 plants-11-02759-f003:**
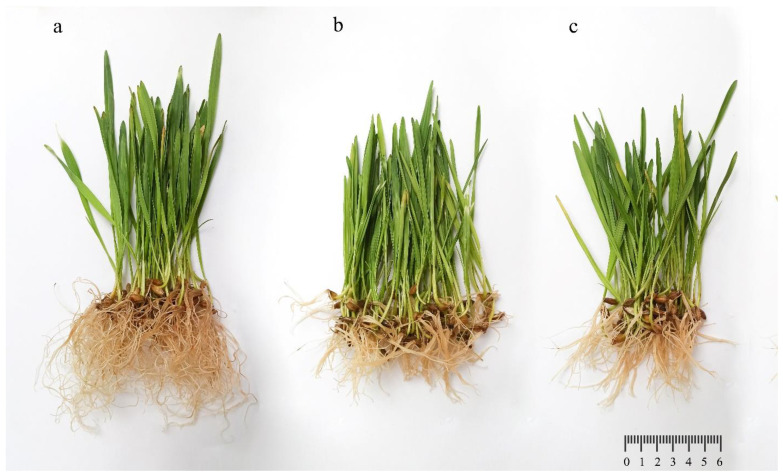
Effects of ZnO NPs on *Hordeum vulgare* L. at tillering phase: (**a**)—control, (**b**)—nano-300, (**c**)—nano-2000.

**Figure 4 plants-11-02759-f004:**
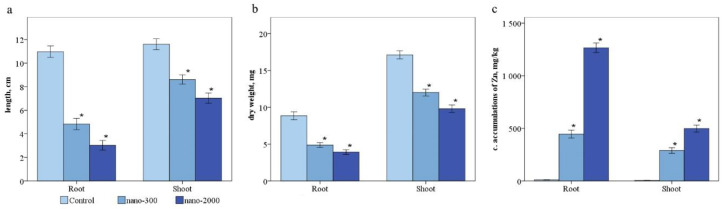
Morphobiometric indices of *Hordeum vulgare* L: (**a**)—shoots and root length, (**b**)—dry weight, (**c**)—Zn content. Statistically significant differences (*p* ≤ 0.05) comparing the treated plants to the control plants are marked with asterix (*).

**Figure 5 plants-11-02759-f005:**
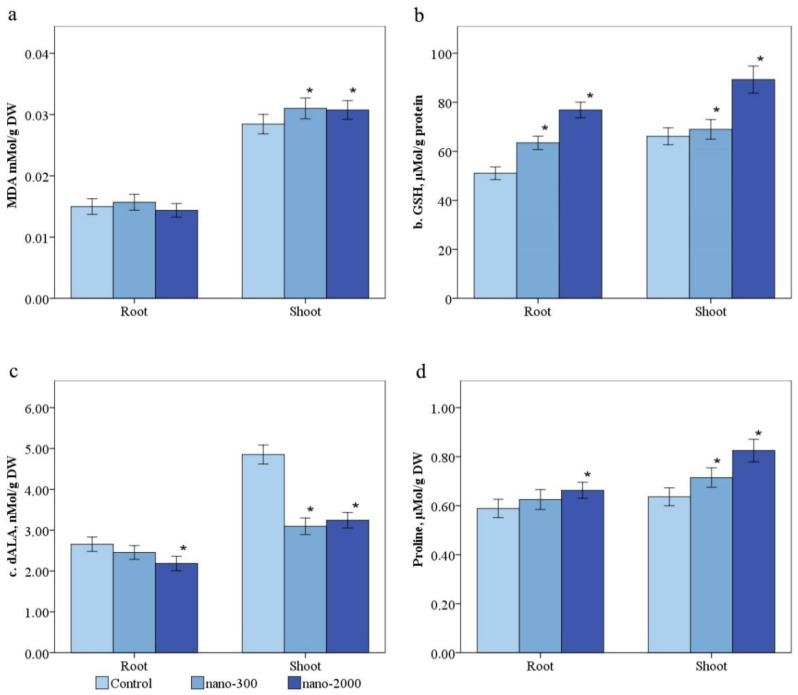
Effect of ZnO-nano on the content of (**a**)—malonic dialdehyde, (**b**)—GSH, (**c**)—δ-aminolevulinic acid, and (**d**)—proline in barley. Statistically significant differences (*p* ≤ 0.05) comparing the treated plants to the control plants are marked with asterix (*).

**Figure 6 plants-11-02759-f006:**
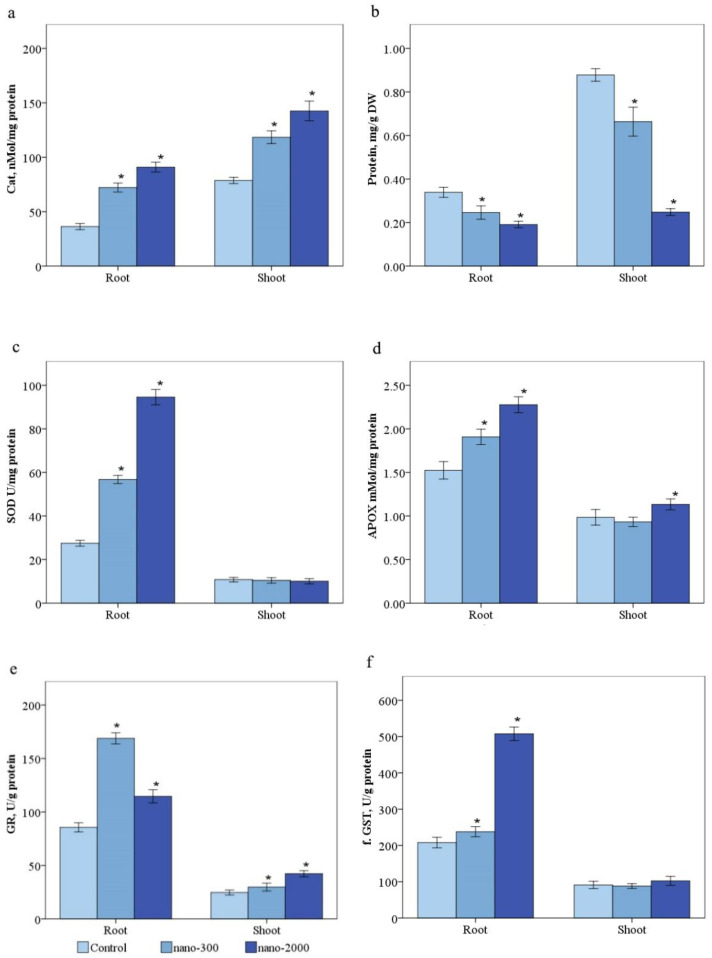
Effect of ZnO-nano on the content of (**a**)—catalase (**b**)—ascorbate-peroxidase, (**c**)—superoxide dismutase, (**d**)—APOX, **e**—glutathione reductase, and **f**—glutathione-s-transferase. Statistically significant differences (*p* ≤ 0.05) comparing the treated plants to the control plants are marked with asterix (*).

## Data Availability

Not applicable.

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
