# Peer review of "Zinc Oxide Nanoparticles: Physiological and Biochemical Responses in Barley (Hordeum vulgare L.)"

_plants, 2022, doi:10.3390/plants11202759_

Round 1
Reviewer 1 Report
Title: Include the scientific name of barley in the title. Zinc Oxide Nanoparticles: Physiological and Biochemical Responses in Barley (Hordeum vulgare L.).
Introduction and conclusions: The novelty of the study is not well established, the authors must specify at the end of the introduction and in the conclusions, what is new about the study.
Introduction and references: There is another study with barley containing cerium nanoparticles to examine the impacts of cerium oxide nanoparticles (nCeO2) on the physiology, productivity and macromolecular composition of barley (Hordeum vulgare L.) that is not in the background, include this work in the introduction and in the references. “Rico, C.M., Barrios, A.C., Tan, W. et al. Physiological and biochemical response of soil-grown barley (Hordeum vulgare L.) to cerium oxide nanoparticles. Environ Sci Pollut Res 22, 10551–10558 (2015). https://doi.org/10.1007/s11356-015-4243-y”
Author Response
Response to Reviewer 1
Dear anonymous reviewer, authors are thankful for your positive evaluation and critical remarks on manuscript (plants-1956023). We have addressed all the comments and supplied track change file where changes are marked. Hope it will be your expectation regarding quality of MS.
- Title: Include the scientific name of barley in the title. Zinc Oxide Nanoparticles: Physiological and Biochemical Responses in Barley (Hordeum vulgare L.).
Answer: We made the necessary corrections in the title.
- Introduction and conclusions: The novelty of the study is not well established, the authors must specify at the end of the introduction and in the conclusions, what is new about the study.
Answer: We made the necessary corrections in the introduction and conclusion.
- Introduction and references: There is another study with barley containing cerium nanoparticles to examine the impacts of cerium oxide nanoparticles (nCeO2) on the physiology, productivity, and macromolecular composition of barley (Hordeum vulgare L.) that is not in the background, include this work in the introduction and in the references. “Rico, C.M., Barrios, A.C., Tan, W. et al. Physiological and biochemical response of soil-grown barley (Hordeum vulgare L.) to cerium oxide nanoparticles. Environ Sci Pollut Res 22, 10551–10558 (2015). https://doi.org/10.1007/s11356-015-4243-y”.
Answer: Thank you for the suggestions. The contents are modified accordingly in the revised manuscript.
Reviewer 2 Report
Title: Zinc Oxide Nanoparticles: Physiological and Biochemical 2 Responses in Barley
The work is very interested and fits the scopes of the journal. I adviced for its publication after several revisions. The most important one being the statistical analysis: none of the statistical information is shown in the figures, it has to be added.
Here are some other comments:
Keywords: instead of the list of all the biochemical parameters measured, I suggest to use the keyword „stress markers“. This way other keywords can be added, related to ZnO NPs.
Introduction
Good
Results
Line 94: the authors should indicate the figure corresponding to the XRD measures.
Figure 1a: the authors need to indicate the mineral structure corresponding to each peak.
Figure 1b: a table indicating the frequency for each particle size will be more visible.
Figure 3: the size of the picture for the control is different from the other (scale smaller).
Figures 4, 5, 6: where is the statistical analysis?
Line 128: where are the data?
Line 152: proline accumulation? Or ALA?
Generally, I suggest to reorganize the presentation of the data, especially regarding the plant stress response. Instead of presenting each molecule separately, I would merge them, for instance, put all the nonenzymatic antioxidants together and all the enzymatic ones together.
Discussion
Line 199: higher than.
Materials and Methods
Line 295: I think a word is missing in the sentence („were collected that are“?).
Line 301: to také into account.
Line 306: which supernatant? The authors did not specify the extraction solution.
For the biochemical analysis, the authors should specify when they used a standard curve and when they used the Beer-Lambert equation.
Author Response
Response to Reviewer 2
Please see the attached file.

Reviewer 3 Report
Line 21 - the activity of SOD increases.
40 - OS is not activated, it develops from a cascade of reactions.
42 - LPO is a process.
43 - MDA is a biochemical marker of oxidative stress.
45 - the accumulation of LPO products - the products are obtained as a result of the oxidation of lipid membranes, proteins, and amino acids
46 - superoxide anion is involved in the development of oxidative stress. Oxylipins are the result of the oxidation of polyunsaturated fatty acids.
69 - SOD catalyzes the dismutation of superoxide ions, therefore it is called superoxide .dismutase, SOD catalyzes the OH radical production from hydrogen peroxide.
86-88 - to pass before 84 - the hypothesis to be before the goal.
It is not clear from the introduction exactly what the role of the zeta potential is and what it means for membrane integrity. Why are you researching it without discussing it?
224 - MDA is a result of free-radical production, it is not a cause! It does not cause ROS but is a product of them.
270 - double check zinc oxide NPS are cofactor of SOD - are they built into the enzyme molecule?
Author Response
Response to Reviewer 3
Authors are thankful for your positive evaluation and critical remarks on manuscript (plants-1956023). We have addressed all the comments and supplied track change file where changes are marked.
- The work is very interested and fits the scopes of the journal. I adviced for its publication after several revisions. The most important one being the statistical analysis: none of the statistical information is shown in the figures, it has to be added.
Answer: We made the necessary corrections.
- Line 21 - the activity of SOD increases.
Answer: We made the necessary corrections.
- Line 40 - OS is not activated, it develops from a cascade of reactions.
Answer: We made the necessary corrections.
- Line 42 - LPO is a process.
Answer: We made the necessary corrections.
- Line 43 - MDA is a biochemical marker of oxidative stress.
Answer: We rewrote the paragraph in the introduction to accommodate the edits.
- Line 45 - the accumulation of LPO products - the products are obtained as a result of the oxidation of lipid membranes, proteins, and amino acids
Answer: We rewrote the paragraph in the introduction accordingly.
- Line 46 - superoxide anion is involved in the development of oxidative stress. Oxylipins are the result of the oxidation of polyunsaturated fatty acids.
Answer: We modified the paragraph in the introduction and deleted the sentence about oxylipins.
- Line 69 - SOD catalyzes the dismutation of superoxide ions, therefore it is called superoxide dismutase, SOD catalyzes the OH radical production from hydrogen peroxide.
Answer: We made the necessary corrections.
- Line 86-88 - to pass before 84 - the hypothesis to be before the goal.
Answer: We made the necessary corrections.
- It is not clear from the introduction exactly what the role of the zeta potential is and what it means for membrane integrity. Why are you researching it without discussing it?
Answer: Thank you for your comment. Unfortunately, the zeta potential did not receive proper attention in the introduction and confused. . A zeta potential result was included as part of the characterization method to demonstrate the possible stability of the colloidal system. As this experiment described only the ZnO NPs used in the experiment, it seems unnecessary to mention it in the introduction. The reviewer's comments prompted us to revise this section to clarify it. We made the following modifications:
“The zeta potential readings of −22 mV for 300 mg/L and −6.5 mV for 2000 mg/L indicated considerable stability. This means that the prepared colloidal system containing ZnO NPs possessed excellent stability and remained intact without agglomeration during the experiment.”
- Line 224 - MDA is a result of free-radical production, it is not a cause! It does not cause ROS but is a product of them.
Answer: We made the necessary corrections.
- Line 270 - double check zinc oxide NPS are cofactor of SOD - are they built into the enzyme molecule? wer #1:
Thank you for your comment. The statement used in the article has been checked, and additional references have been added.
"Cakmak, I. (2000). Tansley Review No. 111 Possible roles of zinc in protecting plant cells from damage by reactive oxygen species. The New Phytologist, 146(2), 185-205".
Round 2
Reviewer 3 Report
No comments